# Investigating the Extracellular-Electron-Transfer Mechanisms and Kinetics of *Shewanella decolorationis* NTOU1 Reducing Graphene Oxide via Lactate Metabolism

**DOI:** 10.3390/bioengineering10030311

**Published:** 2023-03-01

**Authors:** Yu-Xuan Liou, Shiue-Lin Li, Kun-Yi Hsieh, Sin-Jie Li, Li-Jie Hu

**Affiliations:** Department of Environmental Science and Engineering, Tunghai University, Taichung 40704, Taiwan

**Keywords:** extracellular electron transfer, microbial graphene oxide reduction, mediator addition, *Shewanella decolorationis* NTOU1

## Abstract

Microbial graphene oxide reduction is a developing method that serves to reduce both production costs and environmental impact in the synthesis of graphene. This study demonstrates microbial graphene oxide reduction using *Shewanella decolorationis* NTOU1 under neutral and mild conditions (pH = 7, 35 °C, and 1 atm). Graphene oxide (GO) prepared via the modified Hummers’ method is used as the sole solid electron acceptor, and the characteristics of reduced GO (rGO) are investigated. According to electron microscopic images, the surface structure of GO was clearly changed from smooth to wrinkled after reduction, and whole cells were observed to be wrapped by GO/rGO films. Distinctive appendages on the cells, similar to nanowires or flagella, were also observed. With regard to chemical-bonding changes, after a 24-h reaction of 1 mg mL^−1^, GO was reduced to rGO, the C/O increased from 1.4 to 3.0, and the oxygen-containing functional groups of rGO were significantly reduced. During the GO reduction process, the number of *S.* decolorationis NTOU1 cells decreased from 1.65 × 10^8^ to 1.03 × 10^6^ CFU mL^−1^, indicating the bactericide effects of GO/rGO. In experiments adding consistent concentrations of initial bacteria and lactate, it was shown that with the increase of GO additions (0.5–5.0 mg mL^−1^), the first-order reaction rate constants (*k*) of lactate metabolism and acetate production increased accordingly; in experiments adding consistent concentrations of initial bacteria and GO but different lactate levels (1 to 10 mM), the *k* values of lactate metabolism did not change significantly. The test results of adding different electron transfer mediators showed that riboflavin and potassium ferricyanide were able to boost GO reduction, whereas 2,6-dimethoxy-1,4-benzoquinone and 2,6-dimethyl benzoquinone completely eliminated bacterial activity.

## 1. Introduction

Graphene is a promising material that is currently known to have properties such as good mechanical strength and chemical stability, high electron mobility, and excellent thermal and electrical conductivity [1,2]. It is also known as the thinnest, most flexible, and strongest material on Earth, and these unique properties make it well suited for applications in electronics, thermal management, and energy storage and conversion. As a material with potential to be used in the manufacturing of touch screens, solar cells, conductive inks, polymer composites, photodetectors, sensors, nanoelectronic systems, energy storage, and high-frequency demand appliances, graphene has been receiving a great deal of attention in recent years and is seen as remarkably promising [3,4,5,6].

To obtain single-layer, high-quality graphene, methods such as mechanical exfoliation of graphite or chemical vapor deposition methods that select ideal solid catalyst substrates could be applied [4], leading to excessive production costs in large-scale manufacturing. The reduction of graphene oxide (GO) to graphene is considered a popular method due to its low manufacturing cost, high product workability, and its ability to be mass-produced. Since the finished product obtained by this method may be mixed with some multi-layer structures instead of needing a consistent single-layer, this may also indirectly lead to defects in product quality due to the influence of hybrid elements (i.e., not carbon) during chemical reactions or incomplete reduction [1,4], though many of the appealing properties of single-layer graphene are nonetheless preserved. To obtain high-quality (i.e., single-layer) graphene, further separation after GO reduction could be implemented via thermal or plasma processes [7]. Despite the availability of practices for thermal and electrochemical reduction methods [8,9,10,11], the usage of chemical agents to eliminate oxygen functional groups is the most widely applied approach [12]. According to many research projects, at least 50 kinds of reducing agents have been tested [13], including those most commonly used, such as sodium borohydride, lithium aluminum hydride, hydrazine, and others that are quite potent but toxic and harmful, and therefore have a negative impact on the sustainable development of the environment, economy, and society.

To be sustainable, i.e., having “green chemistry”, using biological materials as reducing agents has received great attention. The use of ascorbic acid and amino acids as reducing agents is a typical practical example of this application [14,15], or graphene oxide can be reduced by extracting chemical components from plant leaves and fruit peels [16]. In addition to these environmentally friendly reducing agents, many bacteria can also be used as reducing agents under mild conditions. Among these, the mechanisms of microbial reduction of graphene oxide driven by electrons generated by respiratory metabolism can be roughly divided into two modes: (i) direct electron transfer through direct contact between graphene oxide and surface-associated proteins; and (ii) electrochemically active bacteria that transfer electrons indirectly by secreting redox electron transfer mediators [17,18]. Salas et al. [19] first proposed the reduction of graphene oxide with five different *Shewanella* strains, using lactate as the cultural substrate and keeping the GO reduction occurring under strictly anaerobic conditions. In that study, the *Shewanella* mutants lacking *CymA*, *MtrA*, *MtrB*, *OmcA* and *MtrC* genes were selected to understand which gene is critical, with results revealing that the *MtrA* gene expressing the heme c-type cytochrome in the outer membrane is the key protein in GO reduction. Wang et al. [20] subsequently confirmed that *Shewanella* could reduce GO under aerobic conditions, and the rGO generated in this way showed better electrochemical performance than that generated under anaerobic conditions. In addition, it was further shown that the process of microbial GO reduction does not only depend on the c-type cytochromes on the outer membranes, but also includes the electron transfer mediators secreted by the bacteria. In addition, Jiao et al. [21] also tried to test the mutant strains of *Shewanella oneidensis* MR-1 in the GO-reducing experiments, but contradicting the report of Salas et al. [20], it was pointed out that *CymA* is not dispensable but is as important as cytochromes located on the outer membrane, since the CymA protein could react with externally added chemicals such as riboflavin and anthraquinone-2,6-disulfonate (AQDS) as electron-transfer mediators.

Another typical electrochemically active bacteria that can implement GO reduction is *Geobacter*, which can also use a solid-state electron acceptor (e.g., Fe(III)) for respiration [22]. Kalathil et al. [23] synthesized rGO by using *Geobacter sulfurreducens* in an anaerobic environment under a quiescent state with acetate as its cultural substrate; it is not only evidenced again that the outer membrane c-type cytochrome is important for the GO reduction, but also unexpected elements such as S, P, Fe, Cu, and N were found to be doping on the resultant rGO, thus enhancing the electrocatalytic effect for further application. With regard to the work completed by Lu et al. [24], it was evidenced that *G. sulfurreducens* could promote the secretion of extracellular mediators (e.g., riboflavin) in reaction with high concentrations of GO (≥0.6 mg mL^−1^). Such results also correspond to other studies showing that the *Geobacter* continues to secrete flavin-related substances [25,26], which may first bind to the c-type cytochromes (transcriptions of *OmcB* and *OmcZ*) at the outer cell membrane after secretion to assist with electron transfer.

Inspired by these studies, *Shewanella* was selected as the “reducing agent” to reduce GO and to explore the interaction and correlation between the bacteria and graphene materials. By using the methods of scanning electron microscopy (SEM), transmission electron microscopy (TEM), Fourier transform infrared spectroscopy (FTIR), and X-ray photoelectron spectroscopy (XPS), the surface characteristics of the GO and resultant rGO could be analyzed. In addition, the results of counting the colony-forming units (CFU) show whether microbial GO reduction is growth-associated or not. The results of organic-acid analyses showed that when lactate and GO were selected to be electron donors and acceptors, respectively, changes in the concentrations of the two additions had different degrees of influence on the rates of biochemical metabolisms.

## 2. Materials and Methods

### 2.1. Microorganisms and Cultural Conditions

*Shewanella decolorationis* NTOU1 was obtained from the Bioresource Collection and Research Center in Hsinchu, Taiwan. In brief, prior to inoculation, the precultural process was completed with the LB medium (Luria–Bertani broth containing 10.0 g L^−1^ tryptone, 5 g L^−1^ yeast extract, and 10.0 g L^−1^ NaCl) for 24 h at 35 °C, and the cells were harvested by centrifugation (1000 rpm, 5 min, 25 °C). The obtained biomass was washed three times using a neutral-pH phosphate buffer saline (PBS, 100 mM phosphate and 80 mM KCl, pH 7) and collected in a 50 mL tube for an experiment on graphene oxide reduction to be performed afterward. Cell concentrations at the beginning of graphene-oxide-reduction experiment in the Hungate tube were approximately 0.2–0.3 g L^−1^ (i.e., OD_600_ = 0.4–0.5).

### 2.2. Synthesis of Graphene Oxide

Graphene oxide was prepared using a modified Hummers and Offeman method [27,28]. In a typical procedure, 0.5 g of natural graphite powder was mixed with 46 mL H_2_SO_4_ (98%) and 1 g NaNO_3_ with continuous stirring in an ice bath for 20 min. Subsequently, an entire 5 g of KMnO_4_ was added gently in several separate batches every 10 min, and the mixture was further stirred for 90 min at temperatures ranging from 0–5 °C. The resulting reaction mixture was then transferred to the pre-heated water bath at 35 °C for 90 min of keeping warm. After that, the hot plate was adjusted to raise the temperature to 95 °C and sustain the same level for 30 min, while in the middle, the heating process reached 80 °C, and 46 mL of deionized water was slowly added. After this process was complete, 150 mL of deionized water was added, and the color of the mixture was bright yellow. The resultant mixture could be supplemented by adding 10 mL of 30% hydrogen peroxide to neutralize the unreacted KMnO_4_. The precipitates (i.e., the resulting GO) were repeatedly washed using water under a centrifugal condition (10,000 rpm, 5 min, 20 °C) to completely remove any residual salts and acids. Finally, the GO solution (>10 mg mL^−1^) was diluted using deionized water and stored at room temperature for further experiments. To quantify GO concentration for providing initial conditions, the suspended-solid analysis was applied [29].

### 2.3. Microbial Reduction of Graphene Oxide via S. decolorationis NTOU1

In this study, lactate [30] and GO were used as the sole sources of electron donor and acceptor, respectively. Prior to inoculating bacterial cells, each Hungate tube (total volume = 20 mL) contained lactate medium purged using argon gas to remove trace amounts of oxygen, and 0.5–5.0 mg mL^−1^ of GO (depending on experimental conditions) was added. Moreover, 0.2–0.3 g L^−1^ (OD_600_ = 0.4–0.5) of *S. decolorationis* NTOU1 biomass was inoculated into the GO dispersion. All experiments were performed in duplicate, and the Hungate tubes were incubated at 35 °C in a temperature-controlled and light-blocking oven. To properly maintain a homogeneous condition, a rotating reactor set to 10 degrees of vertical tilt and 15 rpm of rotation was used. After 24 h of incubation, the solids were repeatedly washed using deionized water (10,000 rpm, 5 min, 25 °C) several times, and then air-dried and stored at room temperature as a backup sample for subsequent detection and analysis. The steps to remove residual biological cell samples are similar to those used by Sales et al. [19]. The samples, including those from control experiments, were collected and washed (indicating a process of centrifuge (10,000 rpm, 5 min, 25 °C), removing the supernatant, refilling agent, and resuspension using a vortex) using the following agents in order: pure water (18 MΩ, Millipore Milli-Q Gradient), 75% ethanol, pure water, 1 N HCl, and finally pure water. The rGO samples produced from the experiment with 1.0 mg mL^−1^ GO and 10 mM lactate were then air-dried and investigated by SEM, TEM, FTIR, and XPS analysis.

Furthermore, to understand the effect of additional mediators, 5.0 μM riboflavin, 2.5 mM potassium ferricyanide, 0.2 mM 2,6-Dimethoxy-1,4-benzoquinone (DMoBQ), and 0.2 mM 2, 6-Dimethyl-1,4-benzoquinone (DMBQ) were added to the GO dispersion for the test.

### 2.4. Surface Characterizations and Materials Analyses

An optical microscope (CX31, OLYMPUS, Japan) with an objective lens (PlanC N 100X/1.25, OLYMPUS, Tokyo, Japan) was used, and Capture 2.3 software (The Imaging Source Asia Co., New Taipei City, Taiwan) was subsequently installed to support the AF imaging system (The Imaging Source Asia Co., Taiwan) in order to capture the photos of living bacterial cells and graphene materials. Field emission scanning electron microscopy (SEM, JSM-7401F, JEOL, Japan) and fully automatic digital transmission electron microscopy (TEM, HT7700, HITACHI, Tokyo, Japan) were used to observe the characteristic images of the graphene surfaces and bacteria-graphene interactions. Prior to implementing SEM, the bio-specimens (i.e., GO/rGO pellets with biological cells attached) were fixed in a solution of 2.5% glutaraldehyde and phosphate buffer saline (0.1 M phosphate and 80.0 mM KCl, pH 7.8, PBS) for 16.0 h at 4.0 °C. These specimens were rinsed three times in the PBS to remove the residual glutaraldehyde, for 10.0 min each time. Dehydration was carried out with a series of gradually increasing ethanol concentrations: 50.0, 75.0, 85.0, 95.0, and 100.0%. Finally, the dehydrated specimens were dried using the critical point drier (E3100, Quorum Technologies Ltd., England, UK) and then coated with platinum by ion sputtering (JEC-3000FC, JEOL, Akishima, Tokyo, Japan). As for the bacteria-free GO/rGO specimen, all the aforementioned pre-treatment steps were omitted, except the platinum coating step. With regard to TEM, the GO/rGO pellets were taken out from Hungate tubes after the experiments were completed and were properly diluted using distilled water and then dropped onto the 200-mesh copper grid for further desiccation. To implement the X-ray photoelectron spectroscopy for detecting the composition of elements and covalent bonding on the material surfaces, the samples were air-dried on a gold-plated silicon wafer prior to analyses using a scanning X-ray photoelectron spectroscopy (XPS, ESCALAB 250, Thermo Scientific, London, UK). The X-ray source was an Al cathode ray set at 200 W, with a pass energy of 100 eV (survey scan) and 20 eV (high-resolution scan). In this study, the percentage of elemental composition was calculated based on the prominent elements: C 1s (~287 eV), O 1s (~533 eV), and N 1s (~401 eV) peaks in the survey spectra. The peak area of each element was calculated using ORIGIN PRO 2021b software, and for area correction, the obtained values were divided by using individual atomic sensitivity factors, which are 0.296, 0.711, and 0.477 for C 1s, O 1s, and N1s [31], respectively, while the ratio of carbon and oxygen atoms (C/O) was calculated after the correction step. Multiplex spectra detection was also carried out specifically for the three elements mentioned above to further analyze the composition and changes of molecular bonds. Fourier transform infrared spectra (FTIR) were implemented using a FTIR spectrometer (IR-Prestige-21, Shimadzu, Japan) equipped with an attenuated-total-reflection mode (MIRacle™, PIKE, Madison, WI, USA), which was used to analyze the chemical composition on the surfaces of GO and rGO, with the wavenumber varying from 600 to 4000 cm^−1^. 

### 2.5. Organic-Acid Analyses

Within 24 h of the batch experiment, 1 mL of the mixed liquid from each testing Hungate tube was taken out every 6 h, filtered using a 0.22-μm membrane, and stored at −20 °C for organic-acid analyses. The organic-acid concentration in the mixed liquid was determined in the present study using high-performance liquid chromatography (HPLC). A Shimadzu (Kyoto, Japan) fabrication with a micro-plunger pump (LC-10ADVP), a refractive index detector (RID-10A), and an ICSep COREGEL-87H3 column (7.8 mm × 300 mm) was placed in an oven (CTO-10AVP) at 40 °C and flushed using 0.08 M sulfuric acid as a mobile phase at a flow rate of 0.5 mL min^−1^. To calculate the pseudo-first-order-reaction-rate constant *k* (h^−1^), the changes in the organic-acid concentrations were converted into values after taking a natural logarithm against the time changes, and this indicator was used to compare the differences of biochemical metabolism under various experimental conditions.

### 2.6. Colony Forming Units (CFU)

The mixed-liquid samples were collected at 0, 6, 12, and 24 h of the batch experiment to estimate profiles of cell growth and decay. The biomass quantification using CFU was determined following the protocol of Karas et al. [32]. Principally, each sample was serially diluted in ten-fold increments from 10^0^ to 10^6^. A volume of 100 μL from each dilution was pipetted in triplicates and spread evenly onto separated LB-agar plates. The plates were incubated at 35 °C for 24 h to make the colonies visible. The ultimate CFU was determined by multiplying the counted colony numbers under an adequate dilution ratio (i.e., the condition under which all the colonies are clearly countable without overlap) with one tenth of the dilution number (CFU mL^−1^).

## 3. Results

### 3.1. Characterizations of GO and rGO Interactions with S. decolorationis NTOU1

In this study, we observed the living *S. decolorationis* NTOU1 cells in the process of reducing GO through an optical microscope (OM). As shown in Figure 1, at 0 h, the bacterial cells were clustered around and seemed to have a tendency to approach GO. After 24 h of reaction, well-viable bacterial cells could still be observed attached to the surface or edge of rGO.

SEM images show the morphologies of GO and rGO retaining *S. decolorationis* NTOU1 (Figure 2a,b). It was demonstrably found that after 24 h of reaction, the bacterial cells adhered to the thin graphene films, and the cells were found to have features of depressions and perforation, which were presumably due to the antibacterial effects induced by the sharp edges of the graphene nanosheets [33]. In the images of Figure 2c,d, even without the bacterial cells presenting, the surfaces of the material were observed to change from smooth to wrinkled, and this phenomenon supports the fact that the GO reduction is exactly occurring. Figure 2d shows that the rGO synthesized by *S. decolorationis* NTOU1 consisted of individual sheets closely clumped together, a chemically preferential result due to the reported phenomena that the planar rGO sheets may become thermodynamically stable via bending, corrugating, and wrinkling processes [34].

Nanomaterials with microbes were examined through TEM to obtain an exhaustive microscopic inspection of the morphology. The low-magnification TEM image (4000×, Figure 3a) of *Shewanella*/rGO clearly demonstrated the formation of an interlinked network of rGO with *Shewanella* cells lying on it. Moreover, through image observation at high magnification (10,000×, Figure 3b), characteristic appendages similar to bacterial nanowires [35] or flagellum can also be observed, and their length and diameter were about 2.7 μm and 78.0 nm, respectively.

### 3.2. Elemental and Bond Composition Analysis of Graphene Materials

With regard to the initial GO prepared in this study, as shown in Figure 4, a stretching peak similar to aromatic C=C in GO and rGO was found at ~665 cm^−1^ via FITR, while three characteristic peaks at ~3400 (corresponding to O-H stretching vibration modes of -COOH and C-OH functional groups [36]), ~1630 (corresponding to C=C stretching vibration and O-H bending vibration of water molecules [36,37]), and ~1380 cm^−1^ (corresponding to C-H bending) were also found. Since the carbonyl and carboxyl functional groups are ubiquitous in GO, the signal of ~990 cm^−1^ may be shifted from 1050 cm^−1^, which is corresponding to C-O stretching [38], instead of representing C=C bending [38]. In the FTIR spectrum of rGO, there was a peak of enhanced C=C stretching vibrations around ~1570 cm^−1^, contributed by substituted aromatic rings [36], and the peaks of ~1050 cm^−1^ and 1726 cm^−1^ corresponding to C-O and C=O stretching [37,38], respectively, were found in the rGO spectrum as well, indicating some residual oxygen-containing functional groups left after GO reduction, instead of complete reduction. In addition, after the reduction of GO, the diffraction peaks at 1230 cm^−1^ were N-H bond bending and C-N bond stretching [39], indicating that rGO containing nitrogen was generated by the reduction of *S. decorationis* NTOU1.

XPS was used to assess the degree of reduction of GO using the atomic ratio of carbon and oxygen (C/O) obtained from calculating the ratio of the respective areas under the C 1s and O 1s peaks. The measured sample results showed that the C/O increased from 1.4 to 3.0, indicating that GO was successfully reduced (Figure 5a,b). As shown in Figure 5b, the XPS survey spectrum revealed that clear N signals exist in the rGO samples with the bacterial cells depleted. The atomic percentage of oxygen reduced from 41% for GO to 24% for rGO, and the atomic percentage of carbon increased from 59% for GO to 70% for rGO, indicating that some oxygen-containing functional groups were removed during the reduction process of the bacteria in contact with GO. In addition, the high-resolution C 1s peaks in XPS multiplex spectra were also used to reveal the functional groups on GO and rGO, and for GO, the two main peaks of XPS were C-O at 286.5 eV and C-C at 284.5 eV [40]. When GO was reduced to rGO via *S. decolorationis* NTOU1 metabolism, the C-C peaks became dominant (from 3% to 45%), and the peak proportions of C-O and C=O decreased significantly (from 78% to 51% and from 22% to 29%, respectively, Figure 5c,d). Clearly, N 1s peaks only appeared in rGO compared with GO, which occupied 6% of the three prominent elements (Figure 5b). Figure 5e exhibits that the N 1s spectra of rGO have three broad peaks centered at around 398.8, 400.1, and 401.6 eV [24,39], which are assigned to be pyridinic-N (5%), pyrrolic-N (52%), and graphitic-N (43%), respectively.

### 3.3. Bacterial Growth in the Reaction Process to Generate rGO

In this study, information on cell growth was estimated by recording colony count observations. The natural logarithm of the CFU was plotted as a function of reduction reaction time to represent the results of experiments with and without lactate as the electron donor. As shown in Figure 6, the two conditions had similar linearity, indicating that bacterial cells gradually decreased with the reaction of rGO production, which may be related to the bacteriostatic properties of GO/rGO [33].

### 3.4. Organic Acid Analysis Results under Different Conditions

To separately investigate the effects of electron donor or acceptor on kinetic changes by analyzing the organic-acid variation with the consistent initial cell concentrations maintained at 0.2–0.3 g L^−1^ (ca. OD_600_ = 0.4–0.5), the different experiments were designated as follows: (i) 10 mM of the lactate with different concentrations of GO; (ii) 1 mg mL^−1^ of GO with different concentrations of lactate; and (iii) consistent lactate and GO concentrations with the addition of different electron transporters (i.e., redox compounds, aka mediators). In the 10-mM-lactate experiment, the initial concentrations of GO were controlled to be 0.5, 1.0, 2.5, and 5.0 mg mL^−1^, respectively. From high to low concentrations of GO, the pseudo-first-order reaction rate constants (*k* values) of lactate metabolism were determined to be 0.6, 1.8, 2.4, and 3.0 min^−1^; the *k* values of acetate production were determined to be 2.4, 1.8, 4.2, and 7.2 min ^−1^ in that order, showing that with the increase of different initial GO additions, the rates of lactate metabolism and acetate generation also increased (Figure 7a). In the 1.0-mg mL^−1^ GO experiments, with different initial lactate concentrations (i.e., 1.0, 2.5, 5, 7.5, and 10 mM), the *k* values of lactate metabolism were determined to be 1.56, 0.05, 2.04, 0.36 min^−1^ (1.22 min^−1^ in average), while those of acetate production were found to be 1.56, 0.84, 1.74, 2.28, and 3.72 min^−1^ (2.03 min^−1^ in average), in order; as such, they were found to be oscillating from low to high concentrations of initial lactate without clear correlations (Figure 7f), indicating that the initial GO concentration is the limiting factor governing the entire redox reaction.

In the experiments adding different mediators, results showed that riboflavin and ferricyanide can boost the reduction of GO as evidenced by reading the lactate-metabolism and acetate-production rates, whereas 2,6-Dimethoxy-1,4-benzoquinone (DMoBQ) and 2,6-Dimethyl-1,4-benzoquinone (DMBQ) cannot (Figure 8a,b). In the experiments reducing 1.0 mg mL^−1^ GO, after adding 5 μM of riboflavin, the rates of lactate metabolism and acetate production were boosted from 0.33 and 0.18 mM h^−1^ (no external-mediator addition) to 0.54 and 0.23 mM h^−1^, respectively (ca. 1.6 and 1.3 times amplitude). Further, after adding 2.5 mM ferricyanide, the two rates were boosted to 0.38 and 0.28 mM h^−1^, respectively (ca. 1.2 and 1.5 times amplitude). In the experiments reducing 7.0 mg mL^−1^ GO, after adding 5 μM of riboflavin, the rates of lactate metabolism and acetate production were boosted to be 0.96 and 0.66 mM h^−1^, respectively (ca. 2.9 and 3.7 times amplitude, respectively). After adding 2.5 mM ferricyanide, the two rates were boosted to be 0.60 and 0.36 mM h^−1^, respectively (ca. 1.8 and 1.6 times amplitude, respectively).

In addition, Jiao et al. [21] also tested mutant strains lacking outer membrane cytochromes, finding that even the addition of riboflavin could only rarely contribute to the generation of rGO, an indication that the extracellular electron transfer mechanism of mediators depends on the cooperation of outer membrane c-type cytochromes to be effective.

Furthermore, the aggregation of the reducing GO in the Hungate tubes standing still was observed every 6 h (Figure 9). With 5 μM riboflavin and 2.5 mM ferricyanide externally added, it was noted that black precipitation could be formed with clean supernatant, while with 0.2 mM of DMoBQ and DMBQ externally added, it could be observed that the color of the precipitations remained brown and some materials were not settled perfectly, leaving some turbidity in the supernatant.

## 4. Discussion

It has been reported that during the GO reduction completed by the bacterial cells, the water molecule between the graphene layers can be squeezed out due to the weak π-π interaction [41], which forms a stacking stereo structure that may pack the bacterial cells [42]. In addition to stacking, since the carboxyl groups chiefly located on the GO edges better attract bacteria due to their hydrophilic characteristic, the edges have the priority to be reduced while the interior planes remain hydrophilic, and therefore the curls and wrinkles on the rGO surface are the result, trapping the bacterial cells [24,42]. Such an amphiphilic characteristic for GO has been reported by Cote et al. [43], who demonstrated that GO is capable of creating an emulsifying layer in a mixture of toluene, water, and GO. Another clear demonstration was presented via exhibiting aberration-corrected TEM images showing that the ionizable edges and polyaromatic domains (i.e., hydrophobic) could be coexisting on the same GO sheet [44].

The possibility of extracellular electron transfer (EET) has attracted much attention in the decades since insoluble metal oxide-reducing bacteria were initially reported [45,46]. Mechanisms proposed for EET are known to be mediated by self-secreted reductants such as flavins [47,48], and enzymatic processes mediated by outer membrane proteins [49,50]. Such outer-membrane proteins include cytochrome complexes and may also be involved with EET via conductive appendages (nanowires) produced by *Shewanella* [35], structures that may provide a larger surface area and a possible way for non-motile cells to reach out and attach themselves to the desired solid-state electron acceptors across a long distance. *S. oneidensis* MR-1 nanowires have been reported to be filament bundles measured to be approximately 50–100 nm in diameter and extend tens of microns from the cell surfaces [35,51]. In addition, also in the study completed by Gorby et al. [35], TEM observations revealed the presence of nanocrystalline magnetite forming from ferrihydrite reduction as arranged in linear arrays which depicted features very similar to bacterial nanowires, indirectly confirming that bacterial nanowires serve as the main character in reducing iron mineral. However, in the present study, the TEM images (Figure 3a) exhibit that GO/rGO flakes with large sizes were predominately attached to the cell surface, and only several small pieces attached themselves to the nanowire-like appendage, implying that the mechanisms between mineral and GO reduction may be quite different. While *Shewanella* cells were known to be very motile [52], a special feature found in this study was that the *Shewanella* cells were stopped by the GO and became non-motile (Appendix A). In addition to the cell damage caused by the sharpness of the graphene nanosheets [33], the motional pictures also imply that *Shewanella* cells are able to sense the existence of graphene and therefore perform such a special chemotaxis (i.e., becoming less active).

Consistently, with a higher initial GO concentration (i.e., 7.0 mg mL^−1^), the reaction rates (i.e., lactate metabolism and acetate production) were higher than those with a lower concentration (i.e., 1.0 mg mL^−1^). While adding more GO could be considered to enlarge the contact area of the solid-state electron acceptors, similar to using high-surface-area electrodes [53], the electron-transfer rates could be significantly boosted due to the special physiological characteristics of *Shewanella*. One distinctive observation found in this study that is contradictive to previous electrochemical results [30] is that the riboflavin boosted the electron transfer rates better than the ferricyanide did with 7 mg mL^−1^ GO used as the electron acceptor, even when the concentrations of them were far from each other (i.e., the 5 μM riboflavin against 2.5 mM ferricyanide). This finding can be supported via thermodynamic consideration: the redox potential of rGO was reported to be 44 mV (vs. the standard hydrogen electrode, SHE [24]), which may not favorably receive the electrons from ferrocyanide (i.e., the reduced form of ferricyanide, possessing the redox potential of +300 mV vs. SHE), but may from riboflavin (−300 mV vs. SHE [54]). The flavin compounds, widespread secretions among gammaproteobacteria [48], have been reported as an EET booster in many studies, enhancing current production on microbial electrodes [47] and serving as a crucial role in solid-mineral reduction, which cannot be executed solely by outer-membrane cytochromes [55] in some cases. In one previous study also implementing *Shewanella* GO reduction, when 0.8 mg mL^−1^ GO was supplied and an excess of 12 μM of riboflavin was added, the reduction rate was increased by 2.7 times [21], a greater amplitude than that found in the present study.

The resulting C/O of 3.0 obtained in this study is very similar to the data obtained by Wang et al. (C/O = 3.1, [20]) using *Shewanella* sp. to reduce GO under an aerobic condition after 60 h of reaction, which was achieved in only 24 h under anaerobic conditions. Since our analytical results of FTIR and XPS both showed the heteroatom (i.e., N) doping of rGO after microbial GO reduction, this distinctive effect has been reported in many relevant studies [24,56,57]. With regard to the chemical GO reduction process (i.e., microorganism-free), to implement N doping, the annealing temperature needs to be set at 300–800 °C either under an ammonia atmosphere [58] or by dosing 5-aminotetrazole monohydrate [59] to obtain 5.7% N content in the resulting rGO [58]. Compared with the 5.95% N content obtained in the present study, since no such high temperature is needed, it indicates that microbial GO reduction is a promising technology for heteroatom doping to modify the rGO materials and will be beneficial to enhance the electrochemical reactions for better sensing heavy metal cations [56] and implementing O_2_ reduction on the cathode [57].

## 5. Conclusions

Through a combination of microscopic observation and instrumental analyses, significant changes regarding the stereo structure and chemical characteristics could be seen during the process of microbial GO reduction with *S. decolorationis* NTOU1. The distinctive findings in the present study, which have not yet been presented in relative studies, are extracted as the following conclusions:The number of bacterial cells gradually decreases with reaction time, showing the toxic effect during the GO reduction.While the initial bacteria concentrations were consistent in all the experiments, with the initial lactate concentration fixed, the rates of lactate metabolism and acetate generation increased when the added concentrations of GO increased; with the initial GO concentration fixed, the rates of lactate metabolism did not change significantly. These results indicate that GO concentration is the limiting factor governing the reaction rates of microbial GO reduction.Adding riboflavin and potassium ferricyanide as mediators boosts the generation of rGO, whereas adding DMoBQ and DMBQ eliminates the entire reaction, indicating that not all the redox compounds can help microbial GO reduction.

## Figures and Tables

**Figure 1 bioengineering-10-00311-f001:**
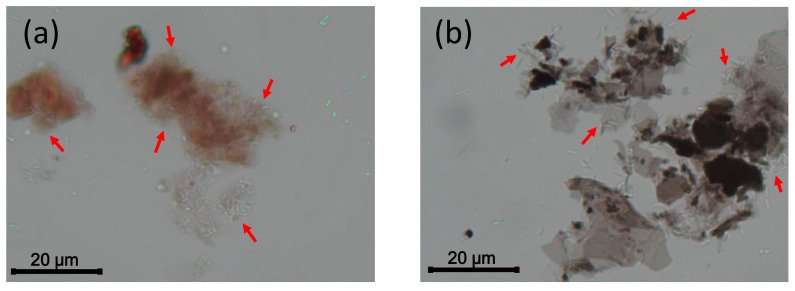
OM images of (**a**) *S. decolorationis* NTOU1 cells with GO at 0 h, and (**b**) *S. decolorationis* cells with rGO after 24 h of reaction (arrows indicate the location of bacterial cells colonizing).

**Figure 2 bioengineering-10-00311-f002:**
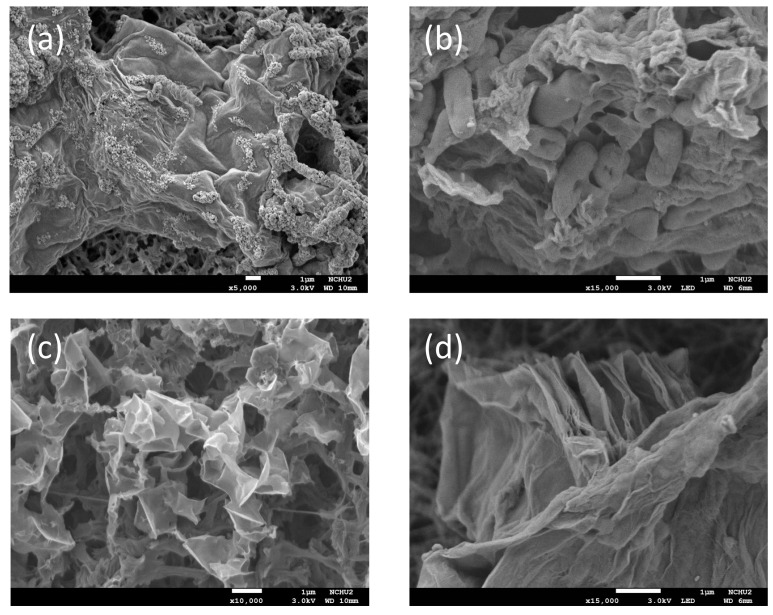
SEM images of (**a**) GO, (**b**) *S. decolorationis* cells with rGO, (**c**) rGO after *S. decolorationis* NTOU1 cells removal, and (**d**) rGO material consisting of individual sheets. It was confirmed that after GO was reduced to rGO by microorganisms, the surface structure changed from flat and smooth to corrugated and wrinkled.

**Figure 3 bioengineering-10-00311-f003:**
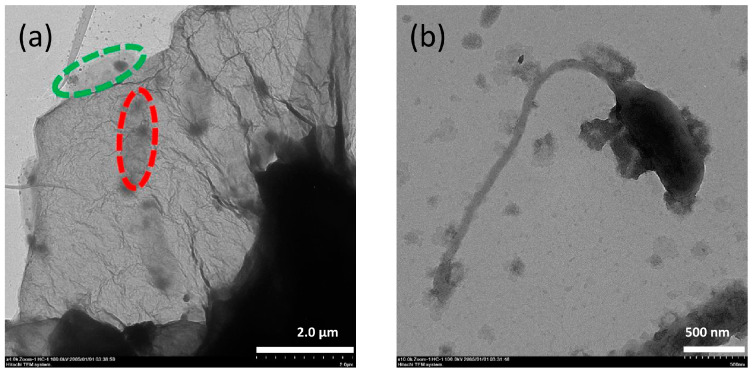
TEM images of (**a**) *S. decolorationis* NTOU1 cells with rGO, and (**b**) a cell with a bacterial-nanowire- or flagellum-like appendage (green, cells locating on the edge; red, cells depositing on the GO/rGO sheet).

**Figure 4 bioengineering-10-00311-f004:**
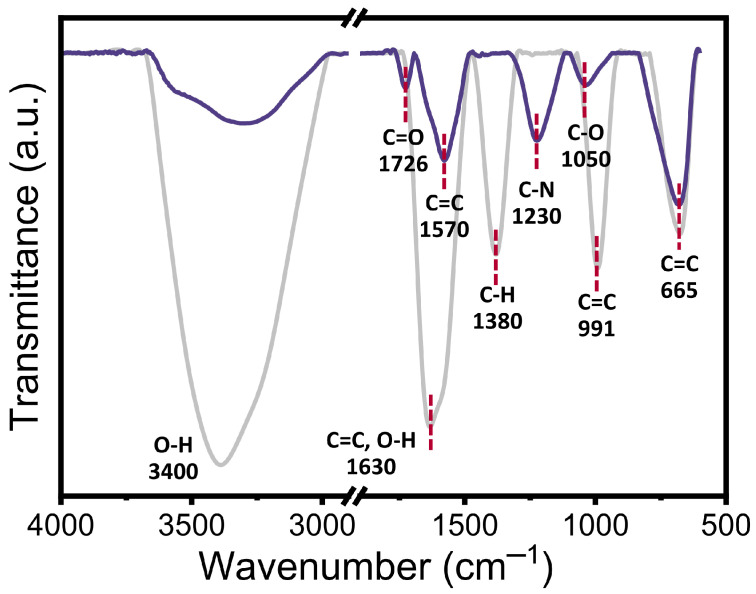
FTIR spectra of GO (gray) and rGO without cells (purple). The peak at ∼1570 cm^−1^ corresponded to C=C stretching emerging after GO reduction. Additionally, the characteristic absorption peak at ~1230 cm^−1^ was detected in rGO corresponding to N-H bending and C-N stretching.

**Figure 5 bioengineering-10-00311-f005:**
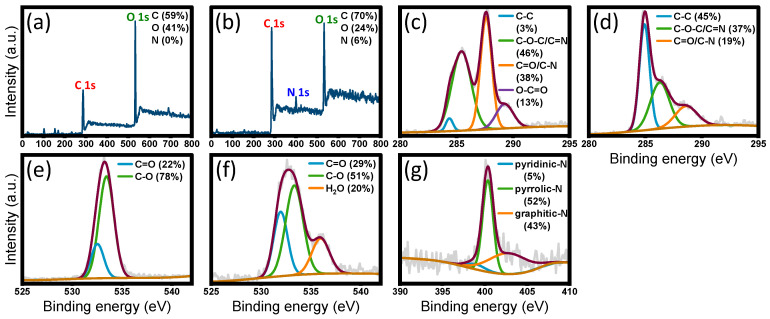
Full-scan XPS surveys for (**a**) GO and (**b**) rGO reduced by *S. decolorationis* NTOU1; XPS C 1s spectra for (**c**) GO and (**d**) rGO reduced by *S. decolorationis* NTOU1; XPS O 1s spectra for (**e**) GO and (**f**) rGO reduced by *S. decolorationis* NTOU1; XPS N 1s spectra for (**g**) rGO reduced by *S. decolorationis* NTOU1. Gray lines, raw data of the experiments; dark red lines, the envelope curve fittings; light brown lines, baselines.

**Figure 6 bioengineering-10-00311-f006:**
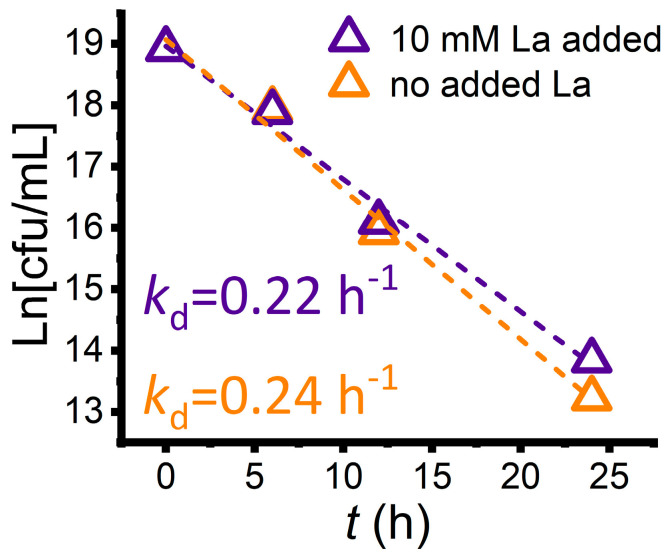
Change of natural logarithmic CFU over time under conditions with/without lactic acid as an electron donor.

**Figure 7 bioengineering-10-00311-f007:**
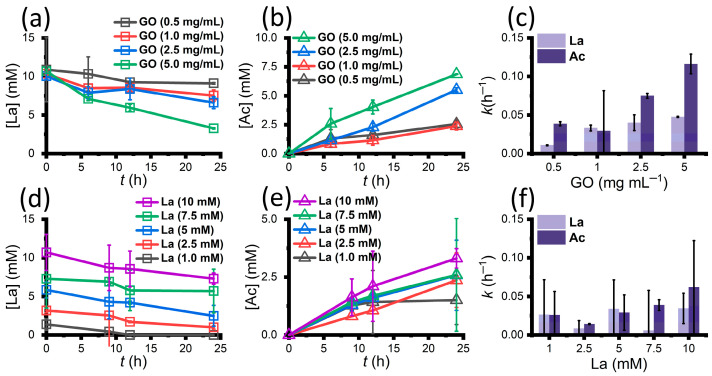
Organic-acid profiles and *k*-value estimations obtained with the controlled initial inoculation (OD_600_ = 0.4–0.5) coupling different experimental conditions, including consistent 10 mM of initial lactate with different GO concentrations (**a**–**c**) and 1.0 mg mL^−1^ of initial GO with different lactate concentrations (**d**–**f**). (**a**,**d**) lactate profiles; (**b**,**e**) acetate profiles; (**c**,**f**) estimated *k* values; GO, graphene oxide; La, lactate; Ac, acetate.

**Figure 8 bioengineering-10-00311-f008:**
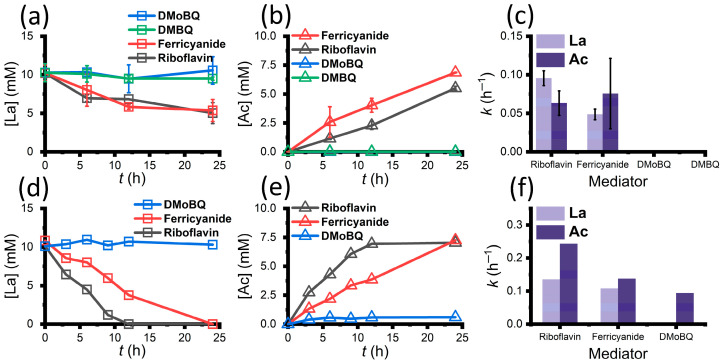
Organic-acid profiles and *k*-value estimations obtained with the controlled initial condition (10 mM lactate and OD_600_ = 0.4–0.5) coupling different experimental conditions, including different concentrations of mediator addition (i.e., 5 μM riboflavin, 2.5 mM ferrycyanide, 0.2 mM DMoBQ, and 0.2 mM DMBQ) with 1.0 mg mL^−1^ (**a**–**c**) and 7.0 mg mL^−1^ (**d**–**f**) of initial GO concentrations. (**a**,**d**) lactate profiles; (**b**,**e**) acetate profiles; (**c**,**f**) estimated *k* values; DMoBQ, 2,6-Dimethoxy-1,4-benzoquinone; DMBQ, 2,6-Dimethyl-1,4-benzoquinone; La, lactate; Ac, acetate.

**Figure 9 bioengineering-10-00311-f009:**
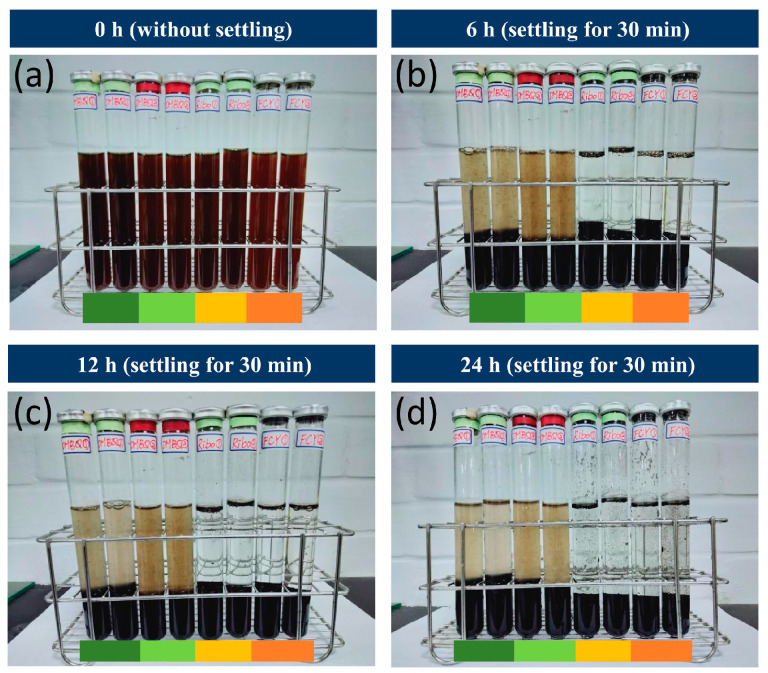
Images photographed under a quiescent condition after different reaction times: (**a**) 0 h; (**b**) 6 h; (**c**) 12 h; and (**d**) 24 h. At 0 h, the image was taken after mixing all the ingredients without settling; at 6, 12, and 24 h, the images were taken with 30 min of settling. The initial concentration of GO was consistently controlled at 1.0 mg mL^−1^. The Hungate tubes of all conditions were prepared as duplicates adding different concentrations of mediators, and were labeled using different colors. From left to right: dark green: 0.2 mM DMoBQ; light green: 0.2 mM DMBQ; yellow: 5 μM riboflavin; orange: 2.5 mM ferrycyanide.

## Data Availability

The data is available with the corresponding author. A part or all of the data can be shared upon reasonable request.

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
