# Peer review of "Investigating the Extracellular-Electron-Transfer Mechanisms and Kinetics of Shewanella decolorationis NTOU1 Reducing Graphene Oxide via Lactate Metabolism"

_bioengineering, 2023, doi:10.3390/bioengineering10030311_

Round 1
Reviewer 1 Report
Congratulations to all authors for their work and preparing this manuscript. There are some comments in this manuscript which should be addressed. Please see the comments below
1. In the abstract, line 12, author should mention the value of normal pressure.
2. In page2, line 88, author should mention electron instead using the symbol e-.
3. Did author ever try to monitor the formation of graphene from graphene oxide by optical density measurement experiment and Raman spectroscopy?
4. Since author mentioned C-O and C=O peak proportions in XPS, author should also mention the disappearance of C=O stretching frequency in the FT-IR spectrum.
5. In page 9, line 318, there is dash (-) in between 10-mM-lactate. What is the reason for this?
6. Also, line 321, check the acetate production values.
7. Similarly in line 324, there is a dash(-) in between 1.0-mg mL-1. What is the reason for this?
8. Author should go through the paragraph 3.4 carefully. Author mentioned wrong figure number. From starting line 324, author mentioned 1.0 mg mL-1 GO experiments, author showed the values of lactate concentration and acetate production values in the lines 326-327. But these values are not matching with the author given figure number.
9. Author should give the explanation for using higher ferricyanide concentration than riboflavin.
10. In line 386, discussion part, author mentioned interior planes remain hydrophilic. Author should confirm this.
11. Author mentioned the electron transfer rates are boosted due to physiological characteristics of Shewanella. Did author try to calculate the electron transfer rate constant?
12. Did author observe aggregation of GO at the higher concentration? Because higher concentration might slower the electron transfer process.
Author Response
Reviewer 1
Congratulations to all authors for their work and preparing this manuscript. There are some comments in this manuscript which should be addressed. Please see the comments below
- In the abstract, line 12, author should mention the value of normal pressure.
- We appreciate your valuable comment. The normal pressure indicates 1 atm in the laboratory, and the actual value will be mentioned in the new manuscript.
- In page2, line 88, author should mention electron instead using the symbol e-.
- We appreciate your valuable comment. We are going to change our wording as “electron acceptor” in the new manuscript.
- Did author ever try to monitor the formation of graphene from graphene oxide by optical density measurement experiment and Raman spectroscopy?
- We appreciate your valuable comment. We chose to use suspended-solid method (Clesceri et al., 2012) to determine the initial concentrations of GO to provide quantified information, instead of using optical-density method. We would also make a clear statement about this analytical method in our new manuscript. Since Raman spectroscopy is a promising method which has the advantage of being able to analyze the water-containing samples and detect the disorder in the carbon lattice (by reading the signal of D band between 1345 cm-1 and 1626 cm-1 of Raman shift, Dresselhaus et al., 2010), our XPS results have revealed that the N atom may be doped, which form a disorder in the resultant rGO. We do agree that the Raman spec may be a powerful tool to clarified the mechanism, but in the present study we did not apply this method.
Reference: Dresselhaus, M.S.; Jorio, A.; Hofmann, M.; Dresselhaus, G.; and Saito, R. Perspectives on carbon nanotubes and graphene raman spectroscopy. Nano Lett. 2010, 10, 751-758.
- Since author mentioned C-O and C=O peak proportions in XPS, author should also mention the disappearance of C=O stretching frequency in the FT-IR spectrum.
- We appreciate your valuable comment. With regard to the initial GO prepared in this study, as shown in Figure 4, a stretching peak similar to aromatic C=C in GO and rGO was found at ~665 cm-1 via FITR, while three characteristic peaks at ~3400 (corresponding to O-H stretching vibration modes of -COOH and C-OH functional groups [36]), ~1630 (corresponding to C=C stretching vibration and O-H bending vibration of water molecules [36,37]), and ~1380 cmï€1 (cor-responding to C-H bending) were also found. Since the carbonyl and carboxyl func-tional groups are ubiquitous in GO, the signal of ~990 cm-1 may be shifted from 1050 cm-1, which is corresponding to C-O stretching, [38]), instead of representing C=C bending [38]. In the FTIR spectrum of rGO, there was peak of enhanced C=C stretching vibrations around ~1570 cm-1 contributed by substituted aromatic rings [36], and the peaks of ~1050 cm-1 and 1726 cm-1 corresponding to C-O and C=O stretching [37,38], respectively, were found in the rGO spectrum as well, indicating some residual oxygen containing functional groups left after GO reduction, instead of complete reduction. In addition, after the reduction of GO, the diffraction peaks at 1230 cm-1 were N-H bond bending and C-N bond stretching [39], indicating that rGO containing nitrogen was generated by the reduction of S. decorationis NTOU1. We would also include this paragraph in our new manuscript, for making a clear statement.
reference:
Kumar, N.; Srivastava, V. C. Simple Synthesis of Large Graphene Oxide Sheets via Electrochemical Method Coupled with Oxidation Process. ACS Omega. 2018, 3, 10233– 10242.
Wei, X. X.; Chen, C. M.; Guo, S. Q.; Guo, F.; Li, X. M.; Wang, X. X.; Cui, H. T.; Zhou, L. F.; Li, W. Advanced visi-ble-light-driven photocatalyst BiOBr–TiO 2–graphene composite with graphene as a nano-filler. J. Mater. Chem. A. 2014, 2, 4667–4675.
IR Spectrum Table & Chart, Sigma Aldrich, available online: https://www.sigmaaldrich.com/TW/en/technical-documents/technical-article/analytical-chemistry/photometry-and-reflectometry/ir-spectrum-table (accessed on 20 January 2023).
Zhang, Z. P.; Rong, M. Z.; Zhang, M. Q. Alkoxyamine with reduced homolysis temperature and its application in repeated autonomous self-healing of stiff polymers. Polym. Chem. 2013, 4, 4648–4654.
5. In page 9, line 318, there is dash (-) in between 10-mM-lactate. What is the reason for this?
- We appreciate your valuable comment. The dashes indicate that these three nouns conjugate together, which could be regarded as an adjective to describe the following object in grammar.
- Also, line 321, check the acetate production values.
- We appreciate your valuable comment. All the digits listed here indicate pseudo-first-order reaction rate constants (k values), and we have confirmed that all the digits are correct. To avoid misunderstanding to the readers, we will replace the wording “the ones” with “the k values”
- Similarly in line 324, there is a dash(-) in between 1.0-mg mL-1. What is the reason for this?
- We appreciate your valuable comment. It shares the same reason aforementioned that we want to describe the “experiment” using a couple of words conjugated together.
- Author should go through the paragraph 3.4 carefully. Author mentioned wrong figure number. From starting line 324, author mentioned 1.0 mg mL-1GO experiments, author showed the values of lactate concentration and acetate production values in the lines 326-327. But these values are not matching with the author given figure number.
- We appreciate your valuable comment. The figure citation at line 330 of our new manuscript should be Figure 7f, instead of Figure 7b. We have made the correction right now.
- Author should give the explanation for using higher ferricyanide concentration than riboflavin.
- We appreciate your valuable comments. The solubility of the mediators is a keynote issue for us to choose the experimental conditions. As for riboflavin, its solubility is ca. 66.0 mg L-1 (i.e., 0.2 mM), and therefore, to make a reliable condition, we considered preparing low levels for the two chemicals rather than concentrations higher than the reported solubilities. Although the solubility of ferricyanide is high, in our previous study we found that Shewanella has a lower affinity to ferricyanide (compared to riboflavin, Li et al., 2018), and eventually the ferricyanide could enhance the EET more when its concentration is high enough (5.5 mM).
reference: Li, S. L.; Yen, J. H.; Kano, K.; Liu, S. M.; Liu, C. L.; Cheng, S. S.; Chen, H. Y. Using metabolic charge production in the tricarboxylic acid cycle (QTCA) to evaluate the extracellular-electron-transfer performances of Shewanella spp. Bioelectrochemistry. 2018, 124, 119–126.
- In line 386, discussion part, author mentioned interior planes remain hydrophilic. Author should confirm this.
- We appreciate your valuable comments. Such an amphiphilic characteristic for GO has been reported by Cote et al. (2010) who evidenced that GO is capable of creating an emulsifying layer in a mixture of toluene, water, and GO. Another clear demonstration was presented via exhibiting aberration-corrected TEM images showing that the ionizable edges and polyaromatic domains (i.e., hydrophobic) could be coexisting on the same GO sheet (Erickson et al., 2010). We would also include this paragraph in our new manuscript, for making a clear statement.
reference:
Cote, L. J.; Kim, J.; Tung, V. C.; Luo, J.; Kim F.; Huang J. Graphene oxide as surfactant sheets. Pure Appl. Chem. 2010, 83, 95-110.
Erickson, K.; Erni, R.; Lee, Z.; Alem, N.; Gannett, W.; Zettl A. Determination of the local chemical structure of graphene oxide and reduced graphene oxide. Adv. Mater. 2010, 22, 4467–4472.
- Author mentioned the electron transfer rates are boosted due to physiological characteristics of Shewanella. Did author try to calculate the electron transfer rate constant?
- We appreciate your valuable comments. In this study, it is very difficult to calculate the electron transfer rate constant since the cell was decaying (cell lysis may deliver electrons as well), and there is no stoichiometry to quantify the electron sink in the reduced graphene oxide. To resolve the question you mentioned, we consider using pure reducing mediator (e.g., reduced riboflavin) reacting with graphene oxide may be a straight way to investigate the electron-transfer rate constant (Ross et al., 2009).
reference: Ross, D. E.; Brantley, S. L.; Tien, M. Kinetic characterization of OmcA and MtrC, terminal reductases involved in respiratory electron transfer for dissimilatory iron reduction in Shewanella oneidensis MR-1. Appl. Environ. Microbiol. 2009, 75, 5218-5226.
- Did author observe aggregation of GO at the higher concentration? Because higher concentration might slower the electron transfer process.
- We appreciate your valuable comments. We do not observe any aggregation after the 24 hours of experiment. Sometimes when we leave the resultant Hungate tube settled for several weeks, we can observe an unfirm rGO aggregation. The attaching figure shows the aggregation of resultant rGO after 1 month of settling.
Reviewer 2 Report
In this manuscript the use of Shewanella to synthetize rGO from GR is reported. The final material has been characterized by SEM, TEM, XPS and FTIR spectroscopy. Manuscript is well organized and structured, the topic is interesting, thus I suggest the publication after some minor revisions:
· the authors reported a band at 665 cm-1 and at 1570 cm-1 relative to the same functional group. This point needs to be revised. In addition, I believe that the decrease of the intensity of bands is not indicative of the reduction. The trasmittance scale is in arbitrary units. The authors should comment the disappearance of bands at ca. 1300 and 1000 cm-1.
· resolution of Figure 5 must be improved
Author Response
In this manuscript the use of Shewanella to synthetize rGO from GR is reported. The final material has been characterized by SEM, TEM, XPS and FTIR spectroscopy. Manuscript is well organized and structured, the topic is interesting, thus I suggest the publication after some minor revisions:
the authors reported a band at 665 cm-1 and at 1570 cm-1 relative to the same functional group. This point needs to be revised. In addition, I believe that the decrease of the intensity of bands is not indicative of the reduction. The trasmittance scale is in arbitrary units. The authors should comment the disappearance of bands at ca. 1300 and 1000 cm-1.
We appreciate your valuable comment. With regard to the initial GO prepared in this study, as shown in Figure 4, a stretching peak similar to aromatic C=C in GO and rGO was found at ~665 cm-1 via FITR, while three characteristic peaks at ~3400 (corresponding to O-H stretching vibration modes of -COOH and C-OH functional groups [36]), ~1630 (corresponding to C=C stretching vibration and O-H bending vibration of water molecules [36,37]), and ~1380 cmï€1 (cor-responding to C-H bending) were also found. Since the carbonyl and carboxyl func-tional groups are ubiquitous in GO, the signal of ~990 cm-1 may be shifted from 1050 cm-1, which is corresponding to C-O stretching, [38]), instead of representing C=C bending [38]. In the FTIR spectrum of rGO, there was peak of enhanced C=C stretching vibrations around ~1570 cm-1 contributed by substituted aromatic rings [36], and the peaks of ~1050 cm-1 and 1726 cm-1 corresponding to C-O and C=O stretching [37,38], respectively, were found in the rGO spectrum as well, indicating some residual oxygen containing functional groups left after GO reduction, instead of complete reduction. In addition, after the reduction of GO, the diffraction peaks at 1230 cm-1 were N-H bond bending and C-N bond stretching [39], indicating that rGO containing nitrogen was generated by the reduction of S. decorationis NTOU1.
We would also include this paragraph in our new manuscript, for making a clear statement.
reference:
36. Kumar, N.; Srivastava, V. C. Simple Synthesis of Large Graphene Oxide Sheets via Electrochemical Method Coupled with Oxidation Process. ACS Omega. 2018, 3, 10233– 10242.
- Wei, X. X.; Chen, C. M.; Guo, S. Q.; Guo, F.; Li, X. M.; Wang, X. X.; Cui, H. T.; Zhou, L. F.; Li, W. Advanced visi-ble-light-driven photocatalyst BiOBr–TiO 2–graphene composite with graphene as a nano-filler. J. Mater. Chem. A. 2014, 2, 4667–4675.
- IR Spectrum Table & Chart, Sigma Aldrich, available online: https://www.sigmaaldrich.com/TW/en/technical-documents/technical-article/analytical-chemistry/photometry-and-reflectometry/ir-spectrum-table (accessed on 20 January 2023).
- Zhang, Z. P.; Rong, M. Z.; Zhang, M. Q. Alkoxyamine with reduced homolysis temperature and its application in repeated autonomous self-healing of stiff polymers. Polym. Chem. 2013, 4, 4648–4654.
resolution of Figure 5 must be improved
We appreciate your valuable comment. We have modified the Figs 4-8 by using bold lines and large words to make the visualization better.
Round 2
Reviewer 1 Report
No comments. Please accept in present form.